# SARS-CoV-2 receptor ACE2 and TMPRSS2 are primarily expressed in bronchial transient secretory cells

Soeren Lukassen[1,2,†] (iD), Robert Lorenz Chua[1,2,†], Timo Trefzer[1,2,†], Nicolas C Kahn[3,4,†], Marc A Schneider[4,5,†], Thomas Muley[4,5], Hauke Winter[4,6], Michael Meister[4,5], Carmen Veith[7], Agnes W Boots[8], Bianca P Hennig[1,2], Michael Kreuter[3,4,*,‡] (iD), Christian Conrad[1,2,**,‡] (iD) & Roland Eils[1,2,9,***,‡] (iD)

## Abstract

The SARS-CoV-2 pandemic affecting the human respiratory system severely challenges public health and urgently demands for increasing our understanding of COVID-19 pathogenesis, especially host factors facilitating virus infection and replication. SARS-CoV-2 was reported to enter cells via binding to ACE2, followed by its priming by TMPRSS2. Here, we investigate *ACE2* and *TMPRSS2* expression levels and their distribution across cell types in lung tissue (twelve donors, 39,778 cells) and in cells derived from subsegmental bronchial branches (four donors, 17,521 cells) by single nuclei and single cell RNA sequencing, respectively. While *TMPRSS2* is strongly expressed in both tissues, in the subsegmental bronchial branches *ACE2* is predominantly expressed in a transient secretory cell type. Interestingly, these transiently differentiating cells show an enrichment for pathways related to RHO GTPase function and viral processes suggesting increased vulnerability for SARS-CoV-2 infection. Our data provide a rich resource for future investigations of COVID-19 infection and pathogenesis.

**Keywords** COVID-19; epithelial differentiation; FURIN; Human Cell Atlas; respiratory tract

**Subject Categories** Chromatin, Transcription & Genomics; Computational Biology; Microbiology, Virology & Host Pathogen Interaction

The EMBO Journal (2020) 39: e105114

## Introduction

In December 2019, a disease affecting predominantly the respiratory system emerged in Wuhan, province Hubei, China, with its outbreak being linked to the Huanan seafood market as about 50% of the first reported cases either worked at or lived close to this market (Chen *et al*, 2020; Huang *et al*, 2020). Among the first 99 reported patients with an average age of 55.5 years, 2/3 were male and 50% suffered from chronic diseases (Chen *et al*, 2020). The disease rapidly spread to other provinces in China, neighboring countries, and eventually worldwide (Wang *et al*, 2020a; Wu & McGoogan, 2020; Zhu *et al*, 2020). The World Health Organization (WHO) named the disease coronavirus disease 2019, COVID-19 (formerly known as 2019-nCov), and the virus causing the infection was designated as severe acute respiratory syndrome coronavirus 2, SARS-CoV-2 (Gorbalenya *et al*, 2020), belonging to the *Coronaviridae* family.

The two coronavirus infections affecting global public health in the 21st century were caused by SARS-CoV and MERS-CoV (Middle East respiratory syndrome coronavirus; de Wit *et al*, 2016). As suggested for SARS-CoV-2, these two coronaviruses had a zoonotic origin with human-to-human transmission after their outbreak (de Wit *et al*, 2016). Deaths by SARS-CoV-2 were solely reported for the critical cases (49%), mainly for patients with preexisting comorbidities such as cardiovascular disease, diabetes, chronic respiratory disease, hypertension, or cancer (Wu &

1  Charité - Universitätsmedizin Berlin, corporate member of Freie Universität Berlin, Humboldt-Universität zu Berlin, and Berlin Institute of Health, Berlin, Germany
2  Center for Digital Health, Berlin Institute of Health (BIH), Berlin, Germany
3  Department of Pneumology and Respiratory Critical Care Medicine, Center for interstitial and rare lung diseases, Thoraxklinik, Heidelberg University Hospital, Heidelberg, Germany
4  Translational Lung Research Center Heidelberg (TLRC), Member of the German Center for Lung Research (DZL), Heidelberg, Germany
5  Translational Research Unit, Thoraxklinik, Heidelberg University Hospital, Heidelberg, Germany
6  Department of Thoracic Surgery, Thoraxklinik, Heidelberg University Hospital, Heidelberg, Germany
7  Division of Redox Regulation, German Cancer Research Center (DKFZ) , Heidelberg, Germany
8  Faculty of Health, Medicine and Life Sciences, Department of Pharmacology and Toxicology, NUTRIM School of Nutrition, Translational Research and Metabolism, Maastricht University, Maastricht, the Netherlands
9  Health Data Science Unit, Heidelberg University Hospital and BioQuant, Heidelberg, Germany
   *Corresponding author. Tel: +49 6221 396 1214; E-mail: kreuter@uni-heidelberg.de
   **Corresponding author. Tel: +49 30 450 543 097; E-mail: christian.conrad@bihealth.de
   ***Corresponding author. Tel: +49 30 450 543 088; E-mail: roland.eils@bihealth.de
   †These authors contributed equally to this work
   ‡These authors contributed equally to this work as senior authors

McGoogan, 2020). Although COVID-19 has a milder clinical impairment compared to SARS and MERS for the vast majority of patients, SARS-CoV-2 infection shows dramatically increased human-to-human transmission rate with the total number of deaths significantly exceeding those of SARS and MERS patients already within the first three months of the COVID-19 outbreak. The emergent global spread of SARS-CoV-2 and its strong impact on public health immediately demands for joint efforts in bio-medical research increasing our understanding of the virus' pathogenesis, its entry into the host's cells, and host factors facili-tating its fast replication that explains the high human-to-human transmission rates.

SARS-CoV-2 is an enveloped virion containing one positive-strand RNA genome, and its sequence has already been reported (Chan *et al*, 2020; Lu *et al*, 2020; Wu *et al*, 2020a,c; Zhou *et al*, 2020). The genome of SARS-CoV-2 comprises 29.9 kb and shares 79.5% and 96% identity with SARS-CoV and bat coronavirus, respectively (Zhou *et al*, 2020). Coronaviruses were reported to use different cellular entry mechanisms in terms of membrane fusion activities after receptor binding (White & Whittaker, 2016). SARS-CoV was previously shown to bind to angiotensin-converting enzyme 2 (ACE2) for cell entry, mediated via the viral surface spike glycoprotein (S protein; Gallagher & Buchmeier, 2001; Li *et al*, 2003; Simmons *et al*, 2013). Comparison of the SARS-CoV and SARS-CoV-2 S protein sequence revealed 76% protein identity (Wu *et al*, 2020c), and recent studies reported that SARS-CoV-2 is also binding to ACE2 *in vitro* (Hoffmann *et al*, 2020; Walls *et al*, 2020; Yan *et al*, 2020; Zhou *et al*, 2020). Subse-quently, the S protein is cleaved by the transmembrane protease serine 2 TMPRSS2 (Hoffmann *et al*, 2020). Simultaneously block-ing TMPRSS2 and the cysteine proteases CATHEPSIN B/L activity inhibits entry of SARS-CoV-2 *in vitro* (Kawase *et al*, 2012), while the SARS-CoV-2 entry was not completely prohibited *in vitro* (Hoffmann *et al*, 2020). It remains to be determined whether additional proteases are involved in priming of the SARS-Cov-2 but not SARS-CoV S protein. One candidate is FURIN as the SARS-CoV-2 S protein contains four redundant FURIN cut sites (PRRA motif) that are absent in SARS-CoV. Indeed, prediction studies suggest efficient cleavage of the SARS-CoV-2 but not SARS-CoV S protein by FURIN (Coutard *et al*, 2020; preprint: Wu *et al*, 2020b). Additionally, FURIN was shown to facilitate virus entry into the cell after receptor binding for several corona-viruses, e.g., MERS-CoV (Burkard *et al*, 2014) but not SARS-CoV (Millet & Whittaker, 2014). While FURIN is generally membrane-bound, an active isoform has been described that can be secreted, potentially facilitating cleavage of the SARS-CoV-2 S protein in the cellular neighborhood (Vidricaire *et al*, 1993). In addition, although *ACE2* was previously described to be expressed in the respiratory tract (Jia *et al*, 2005; Ren *et al*, 2006; preprint: Sung-nak *et al* 2020; Xu *et al*, 2020; preprint: Zhao *et al* 2020; Zou *et al*, 2020), still little is known about the exact cell types expressing *ACE2* and *TMPRSS2* serving as entry point for SARS-CoV and the currently emerging SARS-CoV-2. Therefore, there is an urgent need for investigations of tissues in the upper and lower airways in COVID-19 patients but also healthy individuals to increase our understanding of the host factors facilitating the virus entry and its replication, ultimately leading to treatment strategies of SARS-CoV-2 infections. As pointed out recently

(Zhang *et al*, 2020b), our understanding of host genetic factors involved in COVID-19 disease outcome is still poor.

# Results

## *ACE2* and its co-factor *TMPRSS2* are expressed in the lung and bronchial branches

Here, we established a rich reference dataset that describes the tran-scriptional landscape at the single cell level of the lung and subseg-mental bronchial branches of in total 16 individuals (Fig 1A). Based on this resource, we set out to identify potential key mechanisms likely involved in the SARS-CoV-2 pathway. First, we investigated the expression patterns of the SARS-CoV-2 receptor *ACE2* and the serine protease priming its S protein, *TMPRSS2*, in individual cells in the lung and in subsegmental bronchial branches (Fig 1A).

To quantify gene expression in the lung, single nuclei RNA sequencing was performed on surgical specimens of healthy, non-affected lung tissue from twelve lung adenocarcinoma (LADC) patients, resulting in 39,778 sequenced cell nuclei. All major cell types known to occur in the lung were identified (Fig 1B and C). Independent of the cell types present in the lung, the median *ACE2* levels were below five counts per million (CPM) (Fig 1D), which given a typical mRNA content of 500,000 mRNA molecules per cell indicate that only about half of all cells were statistically expected to contain even a single *ACE2* transcript. The reads per patient and cell type were therefore aggregated into pseudo-bulks, and analysis was continued. As expected from prior literature, the AT2 cells showed highest *ACE2* expression in the lung both in terms of their CPM values (Fig 1D, further referred to as *ACE2*[+] cells, *P*-values in Fig EV1) and the percentage of positive cells (Fig EV2). Further visualizations of expression levels and localiza-tion are shown in Fig EV3. The expression of *TMPRSS2* across cell types of the lung (further referred to as *TMPRSS2*[+] cells) was much stronger with a certain specificity for AT2 cells (Fig 1E, *P*-values in Fig EV1), which is in agreement with other studies (preprint: Sungnak *et al*, 2020).

For the subsegmental bronchial branches, air–liquid interface (ALI) cultures were grown from primary human bronchial epithelial cells (HBECs) and subjected to single cell RNA sequencing, resulting in 17,451 cells across four healthy donors (further referred to as HBECs). In this dataset, we identified all expected cell types, includ-ing recently described and rare cell types such as *FOXN4*-positive cells and ionocytes (Fig 1F, Appendix Fig S1 and S2). Expression levels of *ACE2* were comparable or slightly elevated compared to the lung tissue samples, with the strongest expression being observed in a subset of secretory cells ("secretory3"; Figs 1G and EV1–EV3). Note that due to possible technical differences between single nuclei RNA-Seq (lung) and single cell RNA-Seq (HBECs), the CPM values are not quantitatively comparable. While *TMPRSS2* was less strongly expressed in HBECs than in lung tissue cells, its expres-sion was still markedly higher than that of *ACE2* (Figs 1H and EV1–EV3). Interestingly, the cell population enriched for *ACE2*[+] cells was also the one most strongly expressing *TMPRSS2* (Figs 1G and H, and EV3). In the HBECs, a significant enrichment of the number of *ACE2*[+]/*TMPRSS2*[+] double-positive cells was observed (1.6-fold enrichment, *P* = 0.002, Fig EV4).

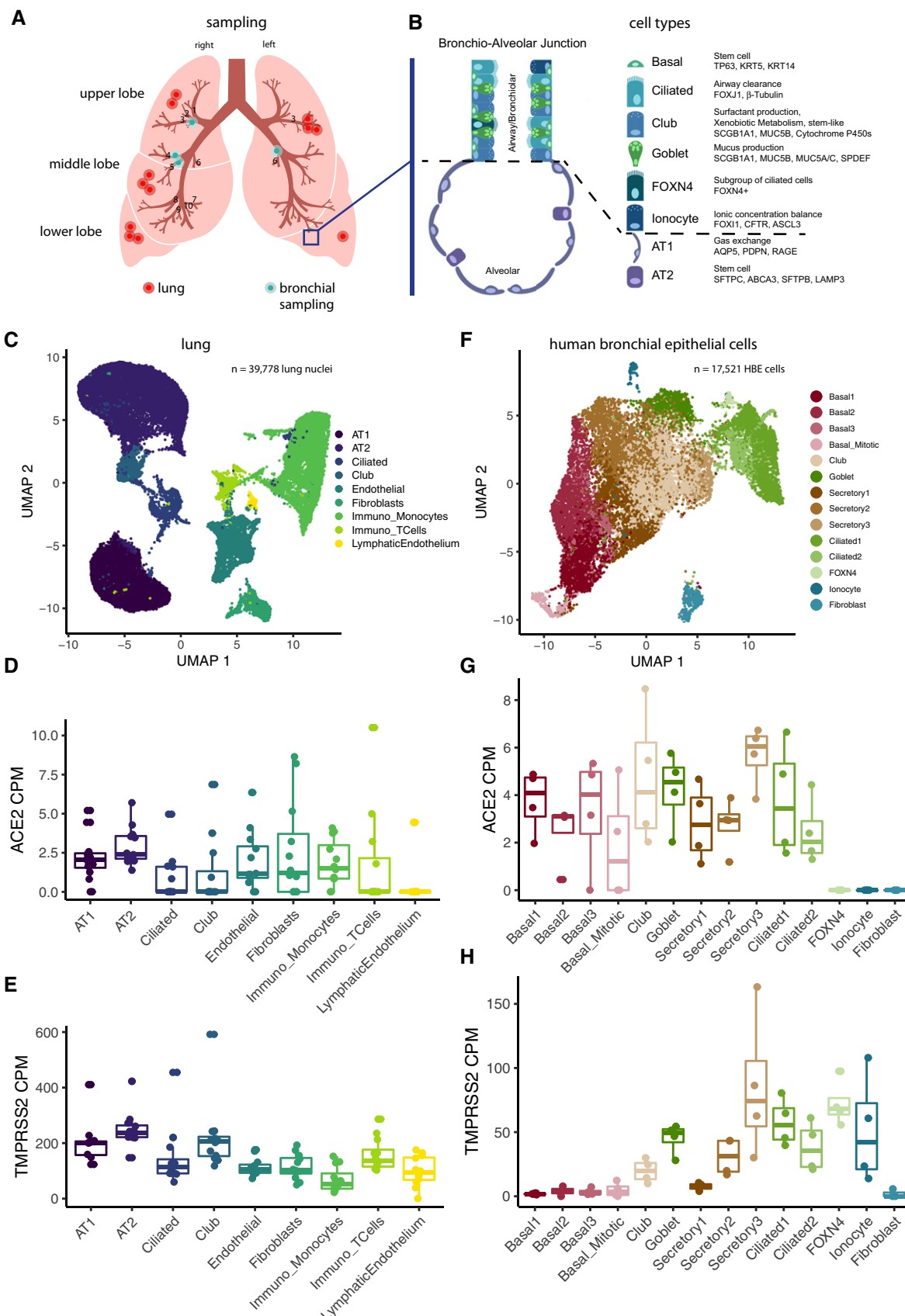

**Figure 1.**

◀

**Figure 1. *ACE2* and *TMPRSS2* are expressed in specific cell types in lungs and HBECs.**

A Sampling location of the surgical lung specimens and human bronchial epithelial cells (HBECs) used in this study. Blue rectangle is zoomed in (B).
B Overview of the major cell types in the lung and airways.
C Uniform manifold approximation and projection (UMAP) of primary lung samples single nuclei RNA sequencing. Cell types are color-coded.
D Expression values of *ACE2* in the cell types of primary lung samples.
E Expression values of *TMPRSS2* in the cell types of primary lung samples.
F UMAP projections of HBEC single cell RNA sequencing data. Cell types are color-coded.
G Expression values of *ACE2* in the cell types of HBECs.
H Expression values of *TMPRSS2* in the cell types of HBECs.

Data information: Boxes in box plots indicate the first and third quartile, with the median shown as horizontal lines. Whiskers extend to 1.5 times the inter-quartile range. Number of patients: Twelve lung samples and four HBEC samples. Each patient is represented as one dot. All individual data points are indicated on the plot.

## Transient secretory cells display a transient cell state between goblet and ciliated cells in HBECs

As the "secretory3" cells in the HBECs showed comparably strong expression of both genes implicated in SARS-CoV-2 entry, with *ACE2* expression levels in a similar to higher range than those in AT2 cells, we decided to further investigate this cell population. As airway epithelial cells are known to continuously renew in a series of differentiation events involving nearly all major HBE cell types in a single trajectory, we mapped the "secretory3" cells according to their position in this hierarchy using reversed graph embedding implemented in Monocle as pseudo-time mapping (Fig 2A; Qiu *et al*, 2017). "Secretory3" cells were found at an intermediate position between club or goblet cells and ciliated cells (Fig 2A), which could be confirmed by shared marker gene expression (Fig 2B). This placed "secretory3" cells at a point shortly before terminal differentiation is reached (Fig 2C and D). Hence, we will refer to these cells as "transient secretory cells".

In agreement with ongoing differentiation toward an epithelial cell lineage, the cell-specific markers of the transient secretory cells showed an enrichment of RHO GTPases and their related pathways (Fig 2E). Note that we find this enrichment of pathways exclusively in the transient secretory cells but in none of the other cell types. As RHO GTPases have been implicated in membrane remodeling and the viral replication cycle, especially entry, replication, and spread (Van den Broeke *et al*, 2014), transient secretory cells may be more permissive to SARS-CoV-2 infection adding to a potential vulnerability caused by considerably high co-expression levels of *ACE2* and *TMPRSS2*. Interestingly, marker genes for transient secretory cells were not enriched in any of the cell types in the whole lung indicating the absence of this transient secretory cell type in the lung samples.

## FURIN-expressing cells overlap with *ACE2*+/*TMPRSS2*+ and *ACE2*+/*TMPRSS2*- cells in lung and in bronchial cells

We next sought to explore whether we could derive additional evidence for other factors recently suggested to be involved in SARS-CoV-2 host cell entry. Therefore, we investigated the expression of *FURIN*, as SARS-CoV-2 but not SARS-CoV was reported to have a FURIN cleavage site in its S protein, potentially increasing its priming upon ACE2 receptor binding. In lung tissue, we detected an overall high expression of *FURIN* (Fig 3A). Across cell types, we observed a marked enrichment of the number of double- and triple-positive cells for any combination of *ACE2*, *TMPRSS2*, and/or *FURIN* expression, indicating a preference for co-expression (Figs 3B, EV2,

and EV3). Interestingly, also transient secretory cells in the HBECs showed an intermediate expression of *FURIN* (Fig 3C). While *TMPRSS2* and *FURIN* were less strongly expressed in HBECs than in distant cells from surgical lung tissue, their expression was still markedly higher compared to *ACE2*. In the HBECs, a significant enrichment of the number of *ACE2*+/*TMPRSS2*+ double-positive cells was observed, while *ACE2*+/*TMPRSS2*+/*FURIN*+ enrichment did not reach significance (Fig 3D). The latter finding, however, may be due to differences in co-expression or the generally lower number of positive cells reducing statistical power. Interestingly, our data showed that the additional possibility of priming the SARS-CoV-2 S protein by FURIN would potentially render 25% more cells vulnerable for infection as compared to by exclusively TMPRSS2 S protein priming. Notably, FURIN was previously reported to be also secreted and active in neighboring cells (Vidricaire *et al*, 1993). Thereby, the expression of FURIN in only some cells in lung tissue or bronchial branches may have a strong impact on the entire cellular neighborhood, dramatically increasing the likelihood for an infection with SARS-CoV-2 of all *ACE2*+ cells in these tissues. However, the presence of the secreted FURIN protease in cells that do not express FURIN remains to be validated by biochemical approaches. Overall, our data suggest that FURIN might increase overall permissiveness of cells in the respiratory tract by potentially equipping more cells with proteolytic activity for SARS-CoV-2 S protein priming after ACE2 receptor binding.

## Sex, age, and history of smoking in correlation with *ACE2* expression

Early epidemiological data on SARS-CoV-2 transmission and spread have suggested age, sex, and history of smoking among others as potential confounding factors impacting SARS-CoV-2 infection (Brussow, 2020; Chen *et al*, 2020; Huang *et al*, 2020; Zhang *et al*, 2020a). We thus investigated these possible risk factors for COVID-19 and their influence on receptor gene expression (Fig 4A). No sex-related difference could be observed in the *ACE2* expression in individual cell types neither in the lung tissue nor in the HBECs (Fig 4C and D). This observation is in full agreement with recent reports based on a large number of COVID-19 patients (Guan *et al*, 2020). Although the first reported cases suggested higher infection rates for males (Chen *et al*, 2020; Huang *et al*, 2020), no significant differences in the infection rate of males and females were found with increasing numbers of COVID-19 patients (Brussow, 2020; Wang *et al*, 2020b; Zhang *et al*, 2020a). It has to be noted, though, that the power of our datasets to detect such changes with lung cells from surgical lung tissue samples from twelve patients (three male,

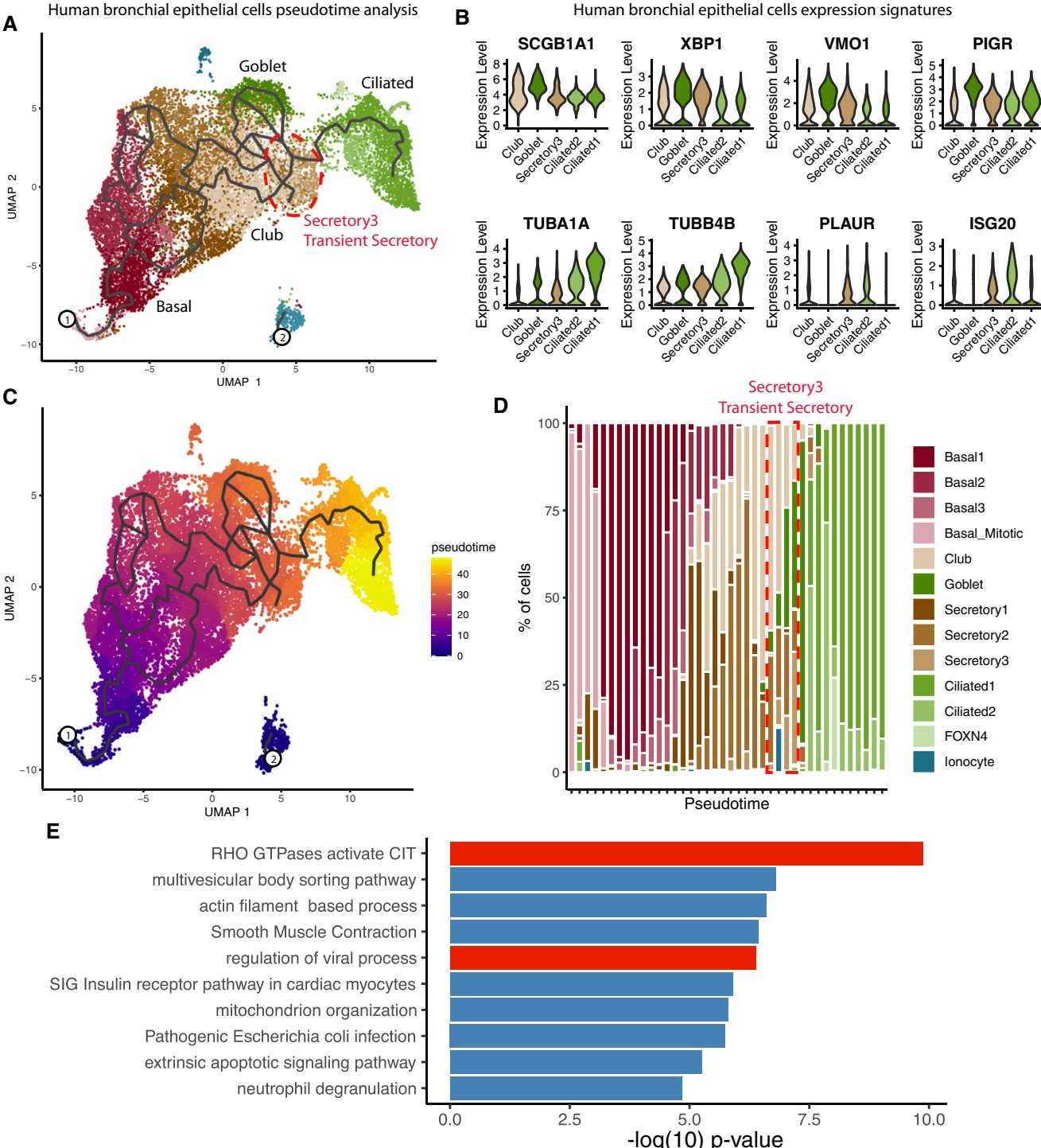

**Figure 2. Characterization of transient secretory (secretory3) cells.**

A   Pseudo-time trajectory projected onto a UMAP embedding of HBECs. The location of the secretory3 cell type is marked by a red outline. Cell types are color-coded. Numbers indicate the beginning of each pseudotime trajectory.

B   Expression of goblet (top row) and ciliated cell markers (bottom row) in club, goblet, secretory3, and ciliated cells.

C   Pseudo-time trajectory projected onto a UMAP embedding of HBECs. Ppseudo-time values are color-coded. Numbers indicate the beginning of each pseudotime trajectory.

D   Cell type composition along a binned pseudo-time axis. The earliest stages are located on the left.

E   Pathway enrichment values for secretory3-specific marker genes.

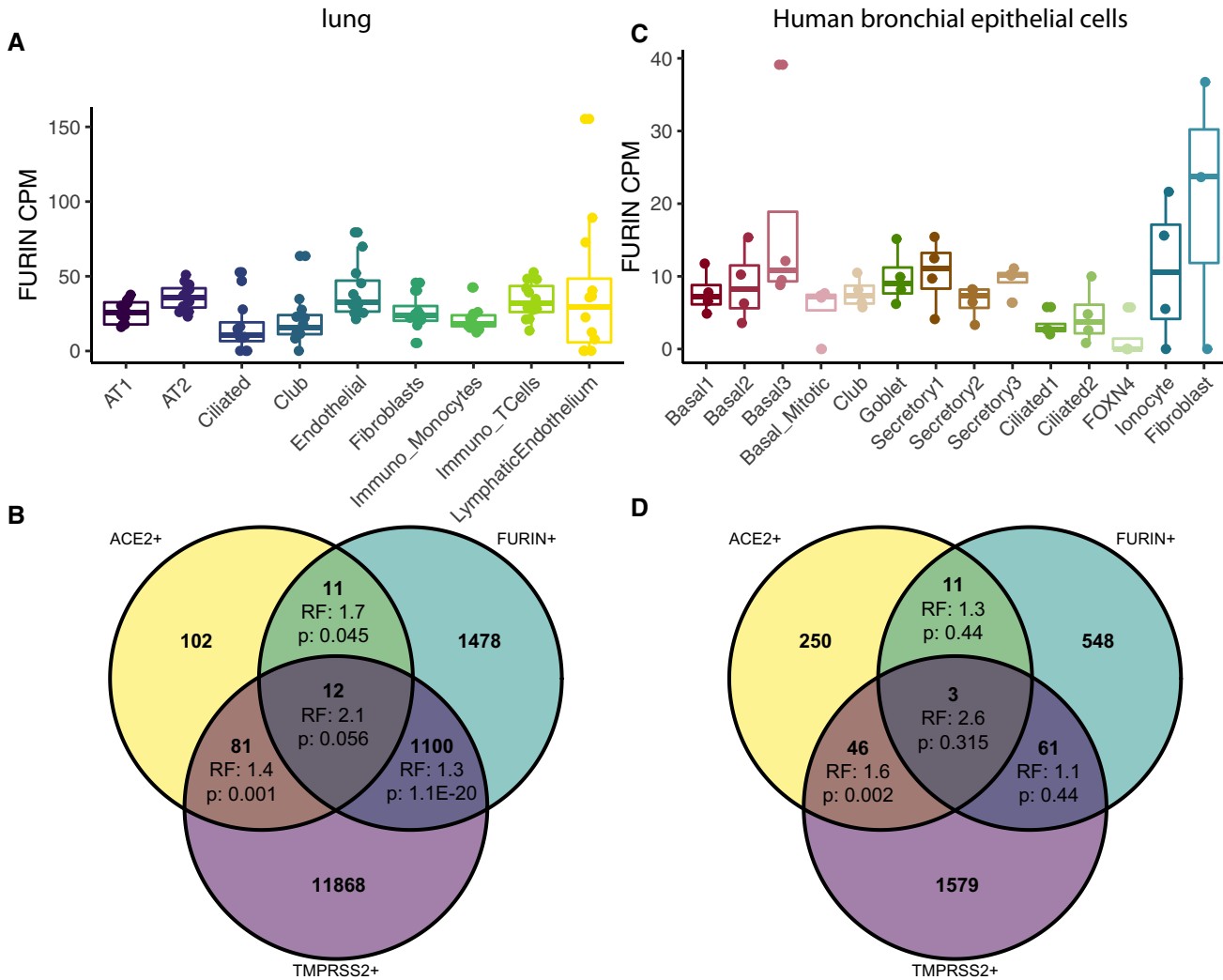

**Figure 3.  *FURIN* is expressed in *ACE2*+ and *ACE2*+/*TMPRSS2*+ cells.**

A   Expression values of *FURIN* in the cell types of primary lung.
B   Overlap of *ACE2*+, *TMPRSS2*+, and *FURIN*+ cells in the lung dataset. RF: representation factor, enrichment. P: hypergeometric tail probability. Total number of cells: 39,778.
C   Expression values of *FURIN* in HBECs.
D   Overlap of *ACE2*+, *TMPRSS2*+, and *FURIN*+ cells in the HBEC dataset. Total number of cells: 17,521.

Data information: RF: representation factor, enrichment. P: hypergeometric tail probability. Total number of cells: 17,451. Boxes in box plots indicate the first and third quartile, with the median shown as horizontal lines. Whiskers extend to 1.5 times the inter-quartile range. Number of patients: Twelve lung samples and four HBEC samples. Each patient is represented as one dot. All individual data points are indicated on the plot.

nine female) and HBECs from four patients (three male, one female) is limited.

On the individual cell type level, no *ACE2* expression differences correlating with age could be observed (Fig 4E and F), and neither did the cell type composition change notably (Fig EV5). Although no dependency of *ACE2* expression on age, gender, and sex was found in single cell populations, we observed interesting trends in the lung for age and gender dependencies when aggregating all reads per sample into a single *ACE2* expression value (Fig 4G and H). Here, we see a trend for increasing overall expression of *ACE2* with age for all female lung samples ($R^2 = 0.40$; $P = 0.065$). Since we only have samples from males of younger age available, we

cannot study age dependencies for males here. However, when comparing *ACE2* expression of all five female and three male samples in the young age group between 40 and 50 years of age, we see a clearly higher overall expression of *ACE2* in males compared to females ($P = 0.048$). Note that smoking did not seem to affect *ACE2* expression levels in the lung cells within our dataset as all smokers in our study were of about the same age and *ACE2* expression correlated with age (Fig 4B).

Altogether, we here demonstrate how single cell sequencing studies of samples from anatomically precisely defined parts of the respiratory system can provide new insights into potential host cell vulnerability for SARS-CoV-2 infection.

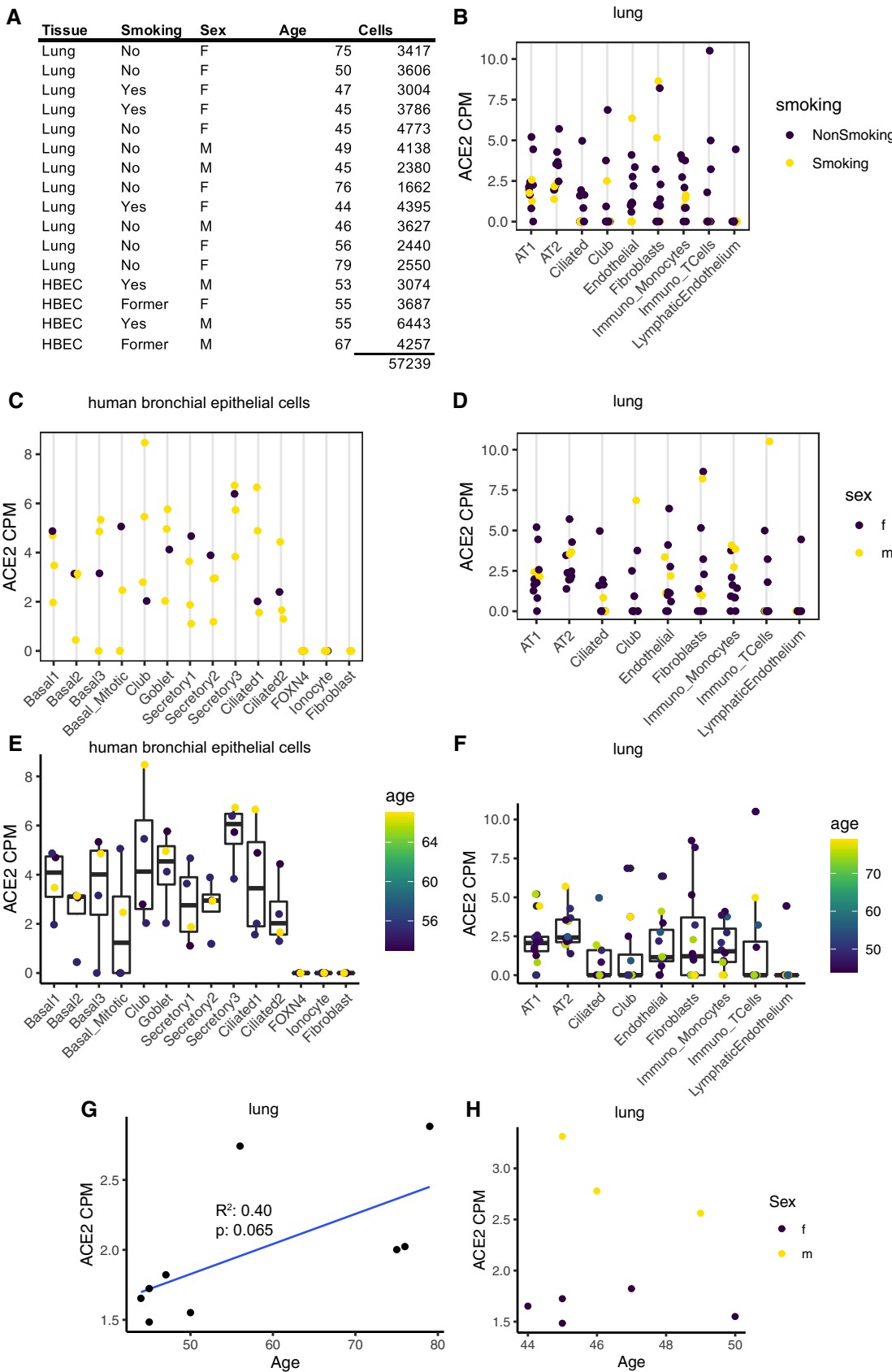

**Figure 4.**

**Figure 4. Age, sex, and smoking behavior as possible factors influencing *ACE2* expression.**

A       Metadata of the sample donors.[†]
B       Expression of *ACE2* in the cell types of the primary lung. History of smoking is color-coded.
C, D    Expression of *ACE2* in HBECs (C) and primary lung (D). Sex is color-coded.
E, F    Expression of *ACE2* in HBECs (E) and primary lung (F). Age is color-coded.
G, H    *ACE2* expression over all cell types versus age in female primary lung cells (G) and primary lung cells of patients aged 40–50 (H). Sex is color-coded.

Boxes in box plots indicate the first and third quartile, with the median shown as horizontal lines. Whiskers extend to 1.5 times the inter-quartile range. All individual data points are indicated on the plot. Nine female patients are shown in (G); eight patients between 40 and 50 years are shown in (H).
[†]Correction added on 28 April 2020, after first online publication: Smoking was corrected to "No" for the fifth donor sample.

## Discussion

The here presented transcriptome data on single cell level of healthy human lung tissues, including surgical lung specimen and subsegmental bronchial branches, contain valuable information about human host factors for SARS-CoV-2 infection and other related infectious diseases affecting the respiratory tract. This resource certainly comes with limitations (see below). We believe the unprecedented depth of our analysis on a single cell level will provide a valuable resource for future mechanistic studies and target mining for pulmonary host factors that are involved in facilitating virus entry and replication, ultimately leading to defining genes that are of urgent interest for studying transcriptional changes in COVID-19 patients or the SARS-CoV-2 pathogenicity in general.

This dataset comprises 16 individuals and a total of 57,299 cells. Thus, this large single cell and nuclei expression resource enables the analysis of weakly expressed genes such as *ACE2* as well as the identification of rare cell types and a transient secretory cell type, for which our data showed a particularly high level of potential vulnerability for SARS-CoV-2 infection, assuming that ACE2 is the receptor that is likely to be crucial for its cell entry. Although we strongly believe that the here presented data are well suited as reference dataset to study SARS-CoV-2 infection on the single cell transcriptional level, we are also well aware of potential limitations of our data.

First, our data are derived from individuals that have no infection history with SARS-CoV-2. However, we deem our data clinically meaningful as our patient cohort is representative for those being affected by SARS-CoV-2-associated disease. The patient cohort analyzed here is in line with recently published data on characteristics of COVID-19 patients with regard to age (this study: median age 48 years—lung cohort or 55 years—HBEC cohort; Chen *et al*, 2020: 55 years; Guan *et al*, 2020: 47 years) (Chen *et al*, 2020; Guan *et al*, 2020; Guan NEJM, Chen Lancet) and especially comorbidity burden (Chen *et al*, 2020) with 50% of the patients being affected by chronic cardiovascular or pulmonary disease or diabetes (this study: 50%; Chen *et al*, 2020: 51%; Guan *et al*, 2020: no data available). There was a different prevalence of smoking or history of smoking in our cohort (this study: 25%—lung cohort or 100%—HBEC cohort; Guan *et al*, 2020: 85%)

Second, we are combining data of different sections of the lung and use two different RNA sequencing methods on single cell level. While single cell RNA sequencing typically delivers higher quality data, the cells must be intact for processing, as damaged cells present a loss of RNA molecules with mitochondrial reads preferentially retained, introducing a skewed expression profile (Luecken & Theis, 2019). This is not observed in single nuclei RNA sequencing, as mitochondria are lost during nuclei isolation. In our HBEC data,

we see a preferential increase in the percentage of mitochondrial reads in certain cell populations rather than across cell types, which may hint toward a physiologically increased energy requirement, for example, due to high levels of protein synthesis in secretory goblet cells (Appendix Fig S3). Adequate sampling of lung specimen for preparing suspensions of intact cells is often challenging. Single nuclei RNA sequencing also reveals transcriptome data on single cell level and can be used on frozen tissue. This method typically provides a more faithful estimate of cell type proportions due to lessened dissociation bias. On the other hand, it typically comes with less reads per cell and less genes detected (Bakken *et al*, 2018), although this bias seems low in our dataset, as the number of both detected genes and unique molecular identifiers (UMIs) is comparable across samples in both datasets (Appendix Fig S3). The identification of closely related cell populations is mostly concordant in both methods, with single cell RNA sequencing being inferior for detection of lowly expressed genes on a single cell level (Bakken *et al*, 2018).

The gene expression shown here for cells from surgical lung tissue samples and ALI-cultured cells derived from the subsegmental bronchial tree (HBECs) should not be directly compared, as the different techniques are likely to have an influence on absolute expression values. But qualitative comparisons of relative abundances as performed here are still appropriate. This is highlighted by the similarity of both datasets in the common QC metrics number of genes per cell and number of UMIs per cell (Appendix Fig S3). Furthermore, the clear separation of clusters in terms of marker genes (Appendix Fig S2) and marked differences in *ACE2* and *TMPRSS2* expression (Fig EV3) indicates that contaminating RNA is not a major issue in either dataset.

Third, the number of samples included here limits the scope of understanding the pathogenicity of SARS-CoV-2 in the context of different confounding factors such as age, gender, and history of smoking. As a consequence of small sample numbers, the significance level for dependency of expression of single genes, most predominantly *ACE2*, on such confounding factors is at best weak. We believe that it is a strength of our data that our clinical samples are well-annotated with respect to anatomical position and patient characteristics. Thus, our data are in principle suited for testing hypothesis derived from larger epidemiological studies. Clearly, measuring gene and protein expression levels for specific genes in larger patient cohorts is required to further substantiate the hypothesis derived from our and others' data.

Lastly, we are aware of the lack of validation on the protein level of the here discussed single cell transcriptomics dataset. However, the here presented data resource enables the scientific community to further investigate hypothesis emerging from our data, in particular as related to potential host factors for SARS-CoV-2 infection.

These hypotheses should be then tested by complementing our data by protein expression and localization data. An important further aspect will be the comparison of our data with transcriptional data obtained from samples of COVID-19 patients.

Bearing the above limitations in mind, we investigated *ACE2* expression levels in all cell types in the lung and bronchial branches revealing very low expression levels that are nevertheless significantly enriched in AT2 cells in the lung as described previously (Hamming *et al*, 2004) and are comparable with the levels in specific cell types in the bronchial branches. As ACE2 is the only currently known SARS-CoV-2 receptor on the host cell surface, we propose that cell populations with comparatively higher *ACE2* expression levels may be more vulnerable for SARS-CoV-2 infection. Future studies both using *in vitro* approaches and COVID-19 patient transcriptome data will be instrumental to test this hypothesis.

The Human Cell Atlas Lung Consortium investigated expression levels of *ACE2* in different tissues based on publicly available single cell sequencing data, including cells of the respiratory tract, such as nasal epithelia and lung tissues (preprint: Sungnak *et al*, 2020). Interestingly, they found *ACE2* to be expressed in various human tissues, including lung, airways, ileum, colon, and kidney. These data may explain the high pathogenic potential of SARS-CoV-2 as described for some critical COVID-19 patients with severe symptoms affecting different organs. In agreement with our data, they also report *ACE2* expression on rather low levels.

While we did not find any dependency of *ACE2* expression on sex, gender, or age on the single cell level, we observed a trend for age dependency on *ACE2* expression aggregated over all cell types in lung samples from females. Recent data suggest that infection rates are similar in children as in adults, but remain undiagnosed since the clinical picture is often subclinical (Bi *et al*, 2020). Hence, it would be of utmost interest to study both healthy and infected children to understand the transcriptional basis accounting for differences in development of clinical symptoms. Furthermore, analyzing further samples from both males and females, both healthy and infected, of different age groups on the single cell level will lay the foundation for understanding vulnerability of different cell types for SARS-CoV-2 infection in different parts of the respiratory system.

One emergent question is why the human-to-human transmission of SARS-CoV-2 is much higher compared to SARS-CoV or MERS-CoV. Potential explanations comprise (i) the binding of SARS-CoV-2 to another, yet unknown receptor on the host cell surface, (ii) enhanced cleavage of the SARS-CoV-2 S protein resulting in higher efficiency of the virus' entry into the cell, and (iii) additional host factors increasing the virus entry into the cell, e.g., by facilitating membrane fusion. We used our reference map and our above described findings, seeking for additional host factors that might be involved in any of those mechanisms.

Coronaviruses were shown to be able to enter into the host cell via several pathways, including endosomal and non-endosomal entry in the presence of proteases (Simmons *et al*, 2004; Matsuyama *et al*, 2005; Wang *et al*, 2008; White & Whittaker, 2016). Different proteases were previously shown to facilitate entry into the cells for different coronaviruses (Hamming *et al*, 2004). Both SARS-CoV and SARS-CoV-2 were shown to share ACE2 as cell surface receptor and TMPRSS2 as the major protease facilitating their entry into the host cell (Hoffmann *et al*, 2020; Walls *et al*, 2020; Yan *et al*, 2020; Zhou

*et al*, 2020). Nevertheless, SARS-CoV-2 human-to-human transmission rates appear to be significantly higher as compared to of SARS-CoV, given the massively higher transmission rate three months after its first detection, with more than 438,000 confirmed infections and close to 20,000 deaths (as compared to 8,096 SARS patients with 2,494 deaths from 2002 to 2003).

SARS-CoV-2 was shown to have a FURIN cleavage site that is absent in SARS-CoV (Coutard *et al*, 2020; preprint: Wu *et al*, 2020b). In addition to the presence of this FURIN cleavage site and experimental evidence that MERS-CoV is primed by FURIN, recent data published by Hoffmann *et al* (2020) suggest an additional protease besides TMPRSS2 and CATHEPSIN B/L as blockage of both completely prohibited the entry of SARS-CoV but not SARS-CoV-2. Further evidence for an involvement of FURIN in the entry of SARS-CoV-2 into the host cell comes from very recent studies predicting an increased binding affinity between the SARS-CoV-2 S protein and the human ACE2 receptor as compared to the SARS-CoV S protein (Wrapp *et al*, 2020; preprint: Wu *et al*, 2020b). Wu *et al* postulate that the cleavage of the S protein at the FURIN cut site might cause the increased binding affinity of SARS-CoV-2 to its receptor, e.g., triggered by structural rearrangements of the cleaved S protein as it was previously shown for other coronavirus S proteins (Kirchdoerfer *et al*, 2016; Walls *et al*, 2016; Wrapp *et al*, 2020). However, the involvement of FURIN in the virus infection pathway remains to be confirmed by future experimental approaches.

As the involvement of an additional protease during S protein priming might enhance entry of SARS-CoV-2 compared to SARS-CoV, we also investigated co-expression of *ACE2*, *TMPRSS2*, and *FURIN*. Using our reference dataset, we were able to show co-expression of *ACE2* with *TMPRSS2* and/or *FURIN* in both healthy human lung tissue and HBECs. The potential additional priming of cells for SARS-CoV-2 infection by FURIN together with evidence from literature that FURIN can be secreted and thereby even act in neighboring cells that do not express FURIN supports the hypothesis for increased human-to-human transmission rates due to the usage of FURIN during virus S protein priming. This hypothesis, however, must be followed up by further studies.

Besides the higher SARS-CoV-2 infection rate of the cells due to FURIN cleavage, the severity of COVID-19 in some of the clinical cases might also be explained by this additional FURIN motif. Recent prediction studies suggest efficient cleavage of the SARS-CoV-2 S protein by FURIN that might result in higher pathogenicity of the virus (Coutard *et al*, 2020; preprint: Wu *et al*, 2020b), potentially due to an increased affinity to the ACE2 receptor (Wrapp *et al*, 2020; preprint: Wu *et al*, 2020b). Further evidence for correlation of increased virus pathogenicity with FURIN priming arises from the observation that the presence of a FURIN-like cleavage site positively correlated with pathogenicity for different influenza strains (Kido *et al*, 2012). It would be of great interest to also investigate FURIN expression in other organs with high *ACE2* expression levels, e.g., in the datasets used by Sungnak *et al* (preprint: Sungnak *et al*, 2020), since those organs have been reported to be associated with severe symptoms in COVID-19 patients.

MERS-CoV is an additional coronavirus that has a FURIN-like motif (Burkard *et al*, 2014). Although this virus did not infect as many individuals as SARS-CoV or the currently emerging SARS-CoV-2, it was highly lethal (37% of MERS patients) in 2016 (WHO

2016, de Wit *et al*, 2016). Although both SARS-CoV-2 and MERS-CoV are likely to be cleaved by FURIN during S protein priming, we speculate that the MERS-CoV receptor DPP4 does not have any or high relevance for COVID-19 as DPP4 is solely expressed in lung cells but not in the subsegmental bronchial branches (Appendix Fig S4). Taken together, these data offer the interesting aspects that both high human-to-human transmission rate of SARS-CoV-2 and the severe COVID-19 cases might be caused by additional cleavage sites, resulting in higher ACE2 binding affinity and/or more efficient membrane fusion.

The most striking finding of this study was the detection of the transient secretory cell population in the bronchial branches expressing *ACE2*. Subsequent cell type marker analysis revealed that these cells reflect differentiating cells on the transition from club or goblet to terminally differentiated ciliated cells, the latter having an important role in facilitating the clearance of viral particles (Sims *et al*, 2008; Dumm *et al*, 2019). RHO GTPase activating CIT and viral processes regulating pathways, i.e., processes related to membrane remodeling or the immune system, are strongly enriched in the transient secretory cells. Interestingly, about 40% of the $ACE2^+$ transient secretory cells are co-expressing the protease *TMPRSS2* that is known to be involved in S protein priming, making this cell type potentially vulnerable for SARS-CoV-2 infection. If we include FURIN as another protease priming in SARS-CoV-2 entry of the host cell, the percentage of transient secretory cells expressing the receptor *ACE2* and either both or one of the proteases *TMPRSS2* and *FURIN* increases in the lung to 50%. Hence, transient secretory cells would be potentially targetable by SARS-CoV-2 infection.

Interestingly, SARS-CoV replication was reported to be 100- to 1,000-fold higher via the non-endosomal pathway as compared to the endosomal pathway, with the non-endosomal pathway being dependent on the presence of proteases (Matsuyama *et al*, 2005). Therefore, our data suggest to experimentally investigate whether SARS-CoV-2 is able to enter into the cell also via the non-endosomal pathway, either by the involvement of several proteases or by the ongoing membrane remodeling in the transient secretory cells, which might include RHO GTPases as one prominent pathway specifically enriched in this cell type.

Taken together, we present a rich resource for studying transcriptional regulation of SARS-CoV-2 infection, which will serve as reference dataset for future studies of primary samples of COVID-19 patients and *in vitro* studies addressing the viral replication cycle. With all limitations in mind, which we discussed above, we demonstrate the potential of this resource for deriving novel and testing existing working hypotheses that need to be followed up by independent studies.

# Materials and Methods

## Reagents and Tools table

| Reagent/Resource | Reference or Source | Identifier or Catalog Number |
|---|---|---|
| **Experimental Models** | | |
| Cryo-conserved lung tissues (human) | Lung Biobank Heidelberg | |
| Air Liquid interface cultures (human) | Lung Biobank Heidelberg | |
| **Recombinant DNA** | | |
| **Antibodies** | | |
| Uteroglobin (CC10) | BioVendor | RD181022220-01 |
| ß-TUBULIN IV | Sigma-Aldrich | T7941 |
| Mucin 5AC | abcam | ab3649 |
| KRT5 | Sigma-Aldrich | HPA059479 |
| **Oligonucleotides and other sequence-based reagents** | | |
| **Chemicals, Enzymes and other reagents** | | |
| DTT | Sigma-Aldrich | D0632 |
| Citric Acid Monohydrate | Sigma-Aldrich | 33114 |
| Sucrose | Sigma-Aldrich | S7903 |
| RNase Inhibitor (40 U/µl) | Takara | 634888 |
| SUPERaseIn™, RNase Inhibitor (20 U/µl) | Thermo Fischer Scientific | AM2694 |
| $MgCl_2$ 1M | Thermo Fischer Scientific | AM9539G |
| KCl 2M | Thermo Fischer Scientific | AM9640G |
| Nuclease free water | Thermo Fischer Scientific | AM9939 |
| Hoechst 33258 | Thermo Fischer Scientific | H3569 |

 *The EMBO Journal*

**Reagents and Tools table** (continued)

| Reagent/Resource | Reference or Source | Identifier or Catalog Number |
|---|---|---|
| Tris buffer solution pH7.5 (1M) Ultrapure Grade | Thermo Fischer Scientific | 15567027 |
| DMEM/F12 | Thermo Fischer Scientific | 11320033 |
| Sodium selenite | Sigma-Aldrich | S5261 |
| Ethanolamine | Sigma-Aldrich | E0135 |
| O-Phosphorylethanolamine | Sigma-Aldrich | P0503 |
| Sodium Pyruvate (100 mM) | Thermo Fischer Scientific | 11360039 |
| Sodium Pyruvate | Gibco | 11360070 |
| Adenine | Sigma | A2786 |
| HEPES | Thermo Fischer Scientific | 15630106 |
| GlutaMAX™ supplement | Thermo Fischer Scientific | 35050038 |
| Rock-inhibitor/ Y-27632 | Stemcell Technologies | 72302 |
| 0.25% Trypsin-EDTA | ThermoFisher Scientific | 25200056 |
| Dispase | Corning | 354235 |
| DPBS | Thermo Fisher Scientific | 14190094 |
| Trypsin EDTA | Thermo Fisher Scientific | 25200056 |
| Dispase | Corning | 354235 |
| BSA | Sigma Aldrich | 05479 |
| PneumaCult-ALI media | StemCell | 05002 |
| PneumaCult-ALI 10x Supplement | StemCell | 05003 |
| Growth Medium SupplementPack | PromoCell | C-39160 |
| **Software** | | |
| Photoshop CS6 | Adobe | |
| Leica Application Suite | Leica | |
| CellRanger 2.1.1 | 10× Genomics (https://github.com/10XGenomics/cellranger/releases/tag/2.1.0) | |
| Seurat 3.0.0 | https://www.nature.com/articles/nbt.4096 (https://github.com/satijalab/seurat) | |
| Monocle 3 | https://www.nature.com/articles/nmeth.4402 (https://github.com/cole-trapnell-lab/monocle3) | |
| Python 3.7.1 | Python Software Foundation (https://www.python.org/downloads/release/python-371/) | |
| Statsmodels 0.9.0 | https://www.statsmodels.org/stable/install.html | |
| Scipy 1.4.1 | https://www.scipy.org/scipylib/download.html | |
| **Other** | | |
| Chromium™ SingleCell 3' Library & Gel Bead Kit v2, 16 rxns | 10× Genomics | PN-120237 |
| Chromium™ SingleCell 3' Library & Gel Bead Kit v2, 4 rxns | 10× Genomics | PN-120267 |
| Chromium™ SingleCell A Chip Kit, 48 rxns | 10× Genomics | PN-120236 |
| Chromium™ SingleCell A Chip Kit, 16 rxns | 10× Genomics | PN-1000009 |
| Chromium™ i7 Multiplex Kit, 96 rxns | 10× Genomics | PN-120262 |
| Dounce homogenizer | Sigma | D8938-1SET |
| Falcon 5 ml Round Bottom Polystyrene Test Tube, with Cell Strainer Snap Cap (35 μm) | Corning | 352235 |
| Countess™ Cell Counting Chamber Slides | Thermo Fischer Scientific | C10314 |
| Countess™ II Automated Cell Counter | Thermo Fischer Scientific | AMQAX1000 |
| ThinCert™ Cell Culture Inserts 0.4 μm 12-well plates | Greiner Bio-One | 665641 |

Reagents and Tools table (continued)

| Reagent/Resource | Reference or Source | Identifier or Catalog Number |
|---|---|---|
| 20 μm Cell Strainer | pluriSelect | 43-50020-03 |
| 35 μm filter | Corning | 352235 |

## Methods and Protocols

All subjects gave their informed consent for inclusion before they participated in the study. The study was conducted in accordance with the Declaration of Helsinki and the Department of Health and Human Services Belmont Report. The use of biomaterial and data for this study was approved by the local ethics committee of the Medical Faculty Heidelberg (S-270/2001 and S-538/2012). Cryopreserved surgical healthy lung tissue from twelve patients was provided by the Lung Biobank Heidelberg. Sex, age, and smoking behavior for every individual are provided in Fig 4A, and additional information can be found in Table 1.

### Lung tissues

Tissue was assembled during routine surgical intervention from lung cancer patients. A representative part of normal lung tissue distant from the tumor was cut into pieces of about 0.5–1 cm³ immediately after resection. Up to 4 pieces were distributed in 2 ml cyrovials (Greiner Bio-One GmbH). Thereafter, the vials were snap-frozen in liquid nitrogen (within 0.5 h after resection). The tissue samples had no direct contact to the liquid nitrogen. After snap-freezing, the vials were stored at −80°C until use (Muley *et al*, 2012). Patients' characteristics are shown in Table 1 (lung tissues).

### Air–liquid interface (ALI) cultures of HBECs

Human bronchial epithelial cells (HBECs) were obtained from endobronchial lining fluid (ELF) by minimally invasive bronchoscopic microsampling (BMS) from subsegmental airways for further investigation of indeterminate pulmonary nodules as described previously (Kahn *et al*, 2009). ELF was obtained from a noninvoled

segment from the contralateral lung. Patients' characteristics are shown in Table 1 (ELF patients).

Primary HBECs were maintained in DMEM/F12 media (Gibco, Carlsbad, CA) supplemented with bovine pituitary extract (0.004 ml/ml), epidermal growth factor (10 ng/ml), insulin (5 μg/ml), hydrocortisone (0.5 μg/ml), triiodo-L-thyronine (6.7 ng/ml) and transferrin (10 μg/ml) (PromoCell), 30 nM sodium selenite (Sigma), 10 μM ethanolamine (Sigma), 10 μM phosphorylethanolamine (Sigma), 0.5 μM sodium pyruvate (Gibco), 0.18 mM adenine (Sigma), 15 mM Hepes (Gibco), and 1x GlutaMAX (Gibco) and in the presence of 10 μM ROCK inhibitor (StemCell). After reaching confluency, cells were transferred to 12-well plates and plated at 90,000 cells/insert using the ThinCert™ with 0.4 μm pores (#665641, Greiner Bio-One). After 2–3 days, cells were airlifted by removing the culture from the apical chamber and PneumaCult-ALI media (StemCell) was added to the basal chamber only. Differentiation into a pseudostratified mucociliary epithelium was achieved after approximately 24–27 days. Afterward, ALI cultures were characterized for their proper differentiation using Uteroglobin/CC10 (#RD181022220-01, BioVendor), Mucin 5AC (#ab3649, Abcam), Keratin 5 (#HPA059479, Merck) and Tubulin beta 4 (#T7941, Merck). To do so, filters were fixed with 4% paraformaldehyde (PFA, Merck) and permeabilized with 0.1% Triton X-100 (Merck). First antibodies were applied over night at 4°C. Filters were incubated with Alexa Fluor 488 and 549 secondary antibodies (Dianova) for 40 min at 37°C. DNA was stained using Hoechst 33342 (#B2261, Merck). Pictures were taken using Leica SP5 confocal microscope and Leica Application Suite software (Leica). Pictures were assembled using Photoshop CS6 (Adobe).

### Single nuclei isolation and RNA sequencing library preparation (lung tissues)

Essentially, single nuclei were isolated as described elsewhere (preprint: Tosti *et al*, 2019). Briefly, snap-frozen healthy lung tissue from lung adenocarcinoma patients was cut into pieces with less than 0.3 cm diameter and single nuclei were isolated at low pH by homogenizing the cells in 1 ml of citric acid-based buffer (Sucrose 0.25 M, Citric Acid 25 mM, Hoechst 33342 1 g/ml) at 4°C using a glass Dounce tissue grinder. After one stroke with the "loose" pestle, the tissue was incubated for 5 minutes on ice, further homogenized with 3–5 strokes of the "loose" pestle, followed by another incubation at 4°C for 5 min. Nuclei were released by five additional strokes with the "tight" pestle, and the nuclei solution was filtered through a 35-μm cell strainer. Cell debris was removed via centrifugation at 4°C for 5 min at 500 $g$, and the supernatant was removed, followed by nuclei cell pellet resuspension in 700 μl citric acid-based buffer and centrifugation at 4°C for 5 min at 500 $g$. After carefully removing the supernatant, the nuclei cell pellet was resuspended in 100 μl cold resuspension buffer (25 mM KCl, 3 mM MgCl$_2$, 50 mM Tris-buffer, 400 U RNaseIn, 1 mM DTT, 400 U

Table 1. Patient characteristics.

| | Surgically lung patients | ELF sampled patients |
|---|---|---|
| Number and gender (male/female) | 12 (3/9) | 4 (3/1) |
| Age (years) | 44–79 (48 ± 13) | 53–67 (55 ± 6.4) |
| ECOG | | |
| 0 | 12 | 3 |
| 1 | 0 | 1 |
| Smoking status | | |
| Current smokers | 3 | 2 |
| Ex-smokers | 0 | 2 |
| Never smokers | 9 | 0 |

ELF, epithelial lining fluid; ECOG, Eastern Cooperative Oncology Group Performance Status. Scale data are expressed as range (mean ± SD).

SUPERase In (AM2694, Thermo Fisher Scientific), 1 g/ml Hoechst (H33342, Thermo Fisher Scientific)). Nuclei concentration was determined using the Countess II FL Automated Cell Counter, and optimal nuclei concentration was obtained by adding additional cold resuspension buffer, if needed. Subsequently, samples were processed using the 10× Chromium device with the 10× Genomics scRNA-Seq protocol v2 to generate cell and gel bead emulsions, followed by reverse transcription, cDNA amplification, and sequencing library preparation following the manufacturers' instructions. Libraries have subsequently been sequenced one sample per lane on HiSeq4000 (Illumina; paired-end 26 × 74 bp)

### Dissociation and single cell RNA sequencing library preparation of ALI cultures

Single cell suspensions were generated by adding 1x DPBS (Thermo, 14190094) to the apical chamber, followed by an incubation at 37°C for 15 min and subsequent washing of the cells with 1× DPBS (three times) to remove excess mucus. Adherent cells were dissociated via 0.25% Trypsin-EDTA (Thermo Fisher Scientific, 25200056) treatment for 5 minutes at 37°C, followed by dispase treatment (Corning, 354235) for 10 min at 37°C. Trypsinization was inactivated by adding cell culture medium with soybean trypsin inhibitor. Cells were spun down at 300 $g$ for 5 min, resuspended in 1× DPBS, and passed through a 20-µm cell strainer (pluriSelect, 41-50000-03) to obtain a single cell solution and to remove cell clumps. Cells were spun at 300 $g$ for 5 min and resuspended in cryopreservation medium (10% DMSO, 10% FBS, and 80% Culture Medium (PneumaCult-ALI) and frozen down gradually before shipping on dry ice.

Cryopreserved cells were thawed at 37°C and spun down at 300×G for 5 min; the cell pellet was resuspended in 1x PBS with 0.05% BSA (Sigma-Aldrich, 05479) and passed through a 35-µm filter (Corning, 352235) to remove cell debris. Single cell suspensions were loaded onto the 10× Chromium device using the 10× Genomics Single Cell 3' Library Kit v2 (10× Genomics; PN-120237, PN-120236, PN-120262) to generate cell and gel bead emulsions, followed by reverse transcription, cDNA amplification, and sequencing library preparation following the manufacturers' instructions. The resulting libraries were sequenced with one sample per lane using the NextSeq500 (Illumina; high-output mode, paired-end 26 × 49 bp) or with two samples per lane using the HiSeq4000 (Illumina; paired-end 26 × 74 bp).

### Pre-processing and data analysis of sequencing libraries

CellRanger software version 2.1.1 (10× Genomics) was used for processing of the raw sequencing data, and the transcripts were aligned to the 10x reference human genome hg19 1.2.0. Low-quality cells were removed during pre-processing using Seurat version 3.0.0 (https://github.com/satijalab/seurat) based on the following criteria: (a) > 200 or, depending on the sample, < 6,000–9,000 genes (surgical lung tissues)/< 3,000–5,000 genes (ALI cultures) and (b) < 15% mitochondrial reads (QC plots in Appendix Fig S3). The remaining data were further processed using Seurat for log-normalization, scaling, merging, clustering, and gene expression analysis. Afterward, all control samples were merged using the "FindIntegrationAnchors" and "IntegrateData" functions with their results being merged again and used for downstream analysis. Monocle3 (Qiu *et al*, 2017; Cao *et al*, 2019) was used to infer cellular trajectories and dynamics. Dimensionality reduction, cell clustering, trajectory graph learning, and pseudo-time measurement through reversed graph embedding were performed with this tool.

### Statistical analyses

Statistical analyses were performed using R and Python 3.7.1 with scipy 0.14.1 and statsmodels 0.9.0. For comparisons of CPM values between two groups, the Wilcoxon rank sum test was used. For multiple-way comparisons, the Kruskal–Wallis test (Kruskal & Wallis, 1952) was used, followed by Dunn's *post hoc* test (Dunn, 1964). To calculate the *P*-value for the overlap between sets, a hypergeometric test was employed. Expected molecule counts per cell were modeled using a binomial distribution and the observed detection probability. Gene set enrichment analysis was performed using Metascape (Zhou *et al*, 2019) on the KEGG, Canonical pathways, GO, Reactome, and CORUM databases. *P*-values were adjusted for multiple testing using the Benjamini–Hochberg method (Benjamini and Hochberg 1995). Boxes in box plots indicate the first and third quartile, with the median shown as horizontal lines. Whiskers extend to 1.5 times the inter-quartile range. All individual data points are indicated on the plot.

### Biomaterial availability

Primary lung and subsegmental bronchial samples are available upon request and upon installment of a material transfer agreement (MTA). Further information and requests for resources and reagents should be directed to and will be fulfilled by the lead contact Roland Eils (roland.eils@charite.de).

## Data availability

There are restrictions to the availability of the dataset due to potential risk of de-identification of pseudonymized RNA sequencing data. Hence, the raw data will be available under controlled access in the following databases:

- Single cell RNA sequencing data at European Genome-Phenome Archive.
- Single nuclei RNA sequencing data at European Genome-Phenome Archive.

Further, our processed data can be accessed and mined through the following resources:

- Count and metadata tables containing patient ID, sex, age, smoking status, cell type, and QC metrics for each cell:

FigShare: https://doi.org/10.6084/m9.figshare.11981034.v1.
Mendeley: https://data.mendeley.com/datasets/7r2cwbw44m/1.
- Mineable single cell gene expression data at: https://digital.bihealth.org/ and at: https://eils-lung.cells.ucsc.edu.

**Expanded View** for this article is available online.

## Acknowledgements

We thank Martin Fallenbüchel and Christa Stolp for collection of tissue samples and Elizabeth Chang Xu for establishment of ALI cultures. We thank Leif Erik Sander, Irina Lehmann, and Saskia Trump for advice in this study.

Cryopreserved surgical lung tissues from patients were kindly provided from the Lung Biobank Heidelberg, a member of the accredited Tissue Bank of the National Center for Tumor Diseases (NCT) Heidelberg, the BioMaterialBank Heidelberg, and the Biobank platform of the German Center for Lung Research (DZL). European Respiratory Society (STRTF - 201804-00377, fellowship CV). This study was supported by the European Commission (ESPACE, 874710, Horizon 2020) and the German Center for Lung Research (DZL, 82DZL00402). This publication is part of the Human Cell Atlas—www.humancellatlas.org/publications.

## Author contributions

NCK, MAS, MM, CC, and RE conceived and designed the project. NCK, MAS, MM, AWB, BPH, MK, CC, and RE supervised the project. RLC, TT, MAS, and CV performed experiments. SL, RLC, TT, and BPH analyzed data. NCK, TM, HW, CV, AWB provided tissue and cells. SL, MAS, TM, BPH, MK, CC, and RE wrote the manuscript, and all authors read, revised, and approved the manuscript.

## Conflict of interest

The authors declare that they have no conflict of interest.

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
