## [Review Process File · The EMBO Journal]

SARS-CoV-2 receptor ACE2 and TMPRSS2 are primarily expressed in bronchial transient secretory cells

Soeren Lukassen, Robert Lorenz Chua, Timo Trefzer, Nicolas C. Kahn, Marc A. Schneider, Thomas Muley, Hauke Winter, Michael Meister, Carmen Veith, Agnes W. Boots, Bianca P. Hennig, Michael Kreuter, Christian Conrad & Roland Eils

Review timeline:

Submission date:	27th Mar 2020
Accepted:	30th Mar 2020

Editor: Hartmut Vodermaier

Transaction Report:

(Please note that this manuscript was previously reviewed at another journal. Since the original reviews are not subject to The EMBO Journal's transparent review process policy, these initial reports and author response to them cannot be published here.)

1st Editorial Decision

30th Mar 2020

Thank you again for submitting/transferring your manuscript together with previous reports and responses from another journal for our editorial consideration. We have now carefully discussed and evaluated everything within our team, and feel that your answers and revisions have satisfactorily addressed the concerns raised to warrant rapid publication in The EMBO Journal, with remaining caveats being adequately emphasized and discussed. Following a number of minor editorial modifications as detailed below, we shall therefore be happy to proceed with acceptance and production of the article.

Corresponding Author Name: Roland Eils (main), Christian Conrad, Michael Kreuter

Manuscript Number: EMBOJ-2020-105114